# In Silico Analysis of Changes in Predicted Metabolic Capabilities of Intestinal Microbiota after Fecal Microbial Transplantation for Treatment of Recurrent *Clostridioides difficile* Infection

**DOI:** 10.3390/microorganisms11041078

**Published:** 2023-04-20

**Authors:** Monica Dahiya, Juan Jovel, Tanya Monaghan, Karen Wong, Wael Elhenawy, Linda Chui, Finlay McAlister, Dina Kao

**Affiliations:** 1Department of Medicine, University of Alberta, Edmonton, AB T6G 2R3, Canada; 2Faculty of Veterinary Medicine, University of Calgary, Calgary, AB T2N 4Z6, Canada; 3National Institute for Health Research, Nottingham Biomedical Research Centre, Nottingham Digestive Diseases Centre, School of Medicine, University of Nottingham, Nottingham NG7 2UH, UK; 4Faculty of Medicine and Dentistry, University of Alberta, Edmonton, AB T6G 2R3, Canada; 5Public Health Laboratory, Alberta Precision Laboratories, Edmonton, AB T6G 2R3, Canada

**Keywords:** fecal microbial transplantation, recurrent *Clostridioides difficile* infection, metabolism, intestinal dysbiosis

## Abstract

Importance: Although highly effective in treating recurrent *Clostridioides difficile* infection (RCDI), the mechanisms of action of fecal microbial transplantation (FMT) are not fully understood. Aim: The aim of this study was to explore microbially derived products or pathways that could contribute to the therapeutic efficacy of FMT. Methods: Stool shotgun metagenomic sequencing data from 18 FMT-treated RCDI patients at 4 points in time were used for the taxonomic and functional profiling of their gut microbiome. The abundance of the KEGG orthology (KO) groups was subjected to univariate linear mixed models to assess the significance of the observed differences between 0 (pre-FMT), 1, 4, and 12 weeks after FMT. Results: Of the 59,987 KO groups identified by shotgun metagenomic sequencing, 27 demonstrated a statistically significant change after FMT. These KO groups are involved in many cellular processes, including iron homeostasis, glycerol metabolism, and arginine regulation, all of which have been implicated to play important roles in bacterial growth and virulence in addition to modulating the intestinal microbial composition. Conclusion: Our findings suggest potential changes in key KO groups post-FMT, which may contribute to FMT efficacy beyond the restored microbial composition/diversity and metabolism of bile acids and short-chain fatty acids. Future larger studies that include a fecal metabolomics analysis combined with animal model validation work are required to further elucidate the molecular mechanisms.

## 1. Introduction

*Clostridioides difficile* infection (CDI) imposes a huge financial and clinical burden to the healthcare system [1,2,3,4]. First-line therapy for the first episode of CDI is oral vancomycin or oral fidaxomicin [5]. Despite appropriate anti-CDI-directed antimicrobial therapy, the rate of recurrence after a primary CDI episode ranges between 10 and 30%, reaching approximately 60% after a third episode [6,7]. Currently, fecal microbial transplantation (FMT) is the most effective treatment for recurrent CDI (RCDI), with efficacy ranging from 60–90% after a single treatment [8,9]. Although FMT is highly effective in RCDI treatment, much remains unknown as to the specific mechanisms mediating its therapeutic effectiveness.

Most CDI cases are triggered by antibiotic use, leading to intestinal dysbiosis. In CDI patients and asymptomatic carriers, microbial richness is reduced in comparison with healthy individuals [10,11]. Specifically, stool samples of CDI patients have shown a significant change in microbial composition and diversity, where the relative abundance of *Bacteroidetes* and *Firmicutes* was found to be significantly reduced whilst the relative abundance of *Proteobacteria* was increased compared with healthy controls [12,13]. Following FMT, intestinal microbial diversity is restored in the recipient, resembling that of the stool donor [14]. This transition from dysbiosis to restored microbial composition and diversity is thought to account for part of the efficacy of FMT. In addition to the restored microbial richness, the production of key microbially derived products, particularly short-chain fatty acids (SCFAs) and secondary bile acids, is thought to contribute to the therapeutic benefits of FMT. The conversion of primary bile acids to secondary bile acids is a microbially dependent process, mediated by bile salt hydrolase (BSH) and 7-α-dehydroxylase, produced solely by the gut microbiome [15,16]. Secondary bile acids inhibit the taurocholate-mediated germination of *C. difficile* spores and can bind to the *C. difficile* TcdB toxin, causing significant conformational changes and preventing the TcdB toxin from binding to the human intestinal surface receptor [17,18]. Moreover, SCFAs play a vital role in maintaining intestinal epithelial integrity and function, protecting against damage caused by bacterial toxins by increasing the tight junction expression, thus reducing intestinal permeability and bacterial translocation [19]. In the gut, SCFA-producing bacteria belong to the *Bacteroidetes* and *Firmicutes* phyla [19]. As a result of intestinal dysbiosis in CDI, there is a decreased relative abundance of SCFA-producing bacteria, leading to a reduced production of SCFAs, potentially contributing to CDI pathogenesis [19].

Whilst host–microbial interactions are complex and incompletely understood [20], other emerging mechanisms of interest include immune-mediated effects on the adaptive and innate immune system, the impact on epigenetic pathways [21], and FMT-mediated changes to serum *N*-glycome. The mucosal immune system is highly regulated by gut microbiota [22]. Essential cellular signaling pathways can be disrupted by the *C. difficile* TcdA- and TcdB-mediated inhibition of Rho-family regulatory GTPases, leading to the disruption of tight junctions and the barrier function [23]. *C. difficile*-associated molecular patterns (PAMPs) interact with mucosal immune cells, leading to the recruitment of phagocytic cells (such as neutrophils and macrophages) and the release of cytokines, several of which have protective effects (e.g., IL-22, IL-1, IL-23, IL-17A, IL-33, and IL-25) versus more damaging effects (e.g., IL-8 and CXCL5) [24]. FMT has been proposed as a mechanism to restore the gut barrier by altering the gut microbial environment and potentially enhance the commensal bacterial production of protective cytokines [24]. 

The impact of FMT on epigenetic pathways and serum *N*-glycosylation profiles may provide a further insight into the therapeutic benefits of FMT. Although there is limited understanding, Monaghan and colleagues demonstrated the upregulation of microRNAs (miRNAs) following FMT in mice models and RCDI patients. Specifically, miR-26b, miR-23a, miR-150, and miR-28-5p were upregulated following successful FMT, leading to a reduced expression of the following inflammatory gene targets: *FGF-21*, *IL-12B*, *IL-18*, and *TNFRSF9* [25]. These changes may confer protective advantages against *C. difficile*-mediated damage to the intestinal epithelium. Additionally, the modulation of the glycosylation profile can impact the function of immunoglobulin G (IgG), influencing its ability to act as either a pro- or anti-inflammatory agent [26]. Complex glycosylation profiles have previously been shown to be associated with pathological conditions such as inflammatory bowel disease (IBD). An exploratory analysis demonstrated that the complexity of serum *N*-glycosylation profiles was reduced following FMT [26]. 

Beyond a restored intestinal microbiome and the metabolism of SCFAs and secondary bile acids, little research to date has explored the role of other microbially derived metabolites following successful FMT. In this study, we attempted to explore other potential processes or pathways that could contribute to the therapeutic benefits.

## 2. Methods

### 2.1. Study Design

We conducted in silico analyses to examine the change in KO groups pre-FMT versus post-FMT by analyzing previously obtained fecal metagenomic sequencing data collected from the original study by Kao and colleagues [27], where RCDI patients were randomized to either oral capsule- or colonoscopy-delivered FMT. Treatment success was defined as the absence of CDI recurrence 12 weeks after FMT. Included patients were adult outpatients between the ages of 18 and 90 years who had experienced at least 3 documented episodes of CDI, each episode occurring within 8 weeks of completing a course of CDI-directed treatment. CDI was confirmed with either: (a) the detection of glutamate dehydrogenase and *C. difficile* toxins A/B (C. diff Quik Chek Complete; Techlab, Blacksburg, VA, USA); or (b) the detection of glutamate dehydrogenase and *C. difficile* cytotoxin B gene (Cepheid), plus the resolution of diarrhea for the current CDI episode. Patients were excluded if they had fulminant CDI; chronic diarrheal illness; inflammatory bowel disease, unless in clinical remission for 3 or more months prior to enrollment; cancer and undergoing therapy; subtotal colectomy, colostomy, or ileostomy; dysphagia; life expectancy of less than 3 months; pregnancy; breastfeeding; and conditions requiring ongoing antibiotic therapy. Serial stool samples were collected before FMT and at weeks 1, 4, and 12 after FMT and frozen at −80 °C. Due to budget constraints, of the 116 recruited patients (mean (SD) age, 58 (19) years; 79 females (68%); 105 completed the trial, with 57 randomized to the capsule group and 59 to the colonoscopy group), 46 (23 oral capsule; 23 colonoscopy) patients had their stool samples analyzed by shotgun metagenomic sequencing. Of the 46 patients, 18 patients had complete sets of samples (i.e., pre-FMT, weeks 1, 4, and 12 post-FMT) and were included for the further analysis in this study. Using the FastDNA Spin Kit for Feces (MP Biomedicals, Santa Ana, CA, USA), stool microbial DNA was extracted for whole-genome shotgun sequencing. Metagenome libraries were created using the Nextera XT platform on a MiSeq instrument (Illumina, San Diego, CA, USA), using a 300 bp paired-end format. Raw sequences were trimmed using a quality score (Q) > 20. Libraries were sequenced at an average depth of ~293,000 paired-ends reads. The taxonomic classification of sequences was conducted with Kraken2 against a customized database that included all full-length genome sequences of bacteria, archaea, viruses, fungi, protozoa from NCBI RefSeq, and the human genome assembly GRCh38. Kraken2 reports the proportion of each library assigned to each taxa so that quantification remains independent of the library size. The sequences generated in this study are publicly available at the SRA portal of NCBI under the accession number SRP117355. A principal coordinate analysis (PCoA) on Bray–Curtis distances, Shannon diversity indices, and Wilcoxon signed-rank tests were computed using Scikit-bio 0.5.1. The permutational multivariate analysis of variance using distance matrices (PERMANOVA) analysis was computed with the function adonis2 from the vegan R package. For functional profiling, sequences with lengths < 100 bp after trimming were discarded. Trimmed reads were aligned against the human genome (GRCh38) and non-aligned reads were subsequently mapped to KO groups using the HUMAnN analysis pipeline [28]. 

Subsequent statistical analyses were completed to determine which KO groups demonstrated a statistically significant change over time. Of those KO groups with a statistically significant change, a literature search using PubMed, Google Scholar, and Ovid Medline databases was performed to propose the potential metabolic pathways involved.

### 2.2. Statistical Analyses

We used univariate linear mixed models to estimate changes in KOs pre-FMT and at weeks 1, 4, and 12 post-FMT, with time (weeks) as the main variable. The time variable assessed if the KO groups changed significantly, estimating the X-unit of change (increase or decrease) per unit of time (weeks). The 95% confidence intervals (CIs) for the change in biomarker per unit/time were calculated. KO groups that demonstrated an overall increase at each time point (weeks 1, 4, and 12 post-FMT) were separated from those that demonstrated an overall decrease. 

Linear mixed model tests were conducted with the R library lmerTest. The Bonferroni correction was used to correct the *p*-values, where 93 KOs with evidence of either an increase or a decrease were assumed to be the number of possible comparisons. A corrected alpha value for a comparison against the *p*-values was calculated as 0.05/93 = 0.0005. Thus, comparisons with a *p*-value smaller than 0.0005 were considered to be significant. All significant comparisons had a fold change > 2.

### 2.3. Ethical Considerations

The study by Kao et al. (2017) [27] received approval from Health Canada (control No. 176567) and the ethic review board at the University of Alberta. The data used in this study were obtained from the original study.

## 3. Results

### 3.1. Patient Demographics

Fecal shotgun metagenomic sequencing data were completed for 46 patients, of which 18 had complete datasets at all time points: pre-FMT and at weeks 1, 4, and 12 post-FMT. Of the 18 patients, the mean (SD) age was 56 (17.1) years; 14 (77.8%) were female and 11 (61.1%) had FMT by colonoscopy.

### 3.2. Changes in Bacterial Taxa Abundance following FMT

Hierarchical clustering (Figure 1A) and principal coordinate analyses (PCoAs) (Figure 1B) showed that, with the exception of a single sample, the effect of FMT on the microbiome taxonomic profile was evident at 1 week post-FMT (1 WPF); after 4 weeks post-FMT (4 WPF), the taxonomic profile of all subjects had substantially changed and remained distinct from its status pre-FMT for at least 12 weeks (12 WPF). The PERMANOVA analysis supported the observed differences and demonstrated that the microbiome of subjects that received FMT was statistically different over time.

### 3.3. Changes in KOs following FMT

Shotgun metagenomic sequencing was mapped to 59,987 KO groups. Of these, 93 KO groups demonstrated either an increase or a decrease post-FMT and underwent a further analysis to determine if the change was statistically significant. The statistical significance was determined by the presence of *p*-values less than the corrected *p*-value, based on the Bonferroni correction (*p*-value < 0.0005). Of the 93 KO groups, 2 demonstrated a statistically significant increase post-FMT (Table 1), whereas 24 demonstrated a statistically significant decrease post-FMT (Table 2).

Of the KO groups that increased after FMT, one was involved in DNA replication (K02315), whereas the other was involved in signaling pathways in response to environmental stimuli (K07646). KO groups with an observed decrease post-FMT were implicated in several cellular and molecular processes, including metabolism (9/24), DNA synthesis/replication/repair (7/24), cellular signaling (1/24), substrate transport (6/24), and other miscellaneous functions (1/24).

## 4. Discussion

We demonstrated the changes in KO groups involved in a variety of cellular processes (metabolism, DNA replication, cellular signaling, and substrate transport), the majority of which were housekeeping functions. Despite being involved in mainly housekeeping functions, the changes we observed following successful FMT were of clinical significance because RCDI-associated intestinal dysbiosis may suppress or promote these functions, which normalize once microbial diversity is restored. In particular, the pathways of interest included iron homeostasis, glycerol metabolism, and arginine regulation.

We observed a decrease in KO group K03402 (transcriptional regulator of arginine metabolism). The transcriptional regulator ArgR inhibits the transcription of multiple genes involved in the biosynthesis and uptake of arginine [54,55]. Additionally, ArgR regulates arginine catabolism by activating the *astCADBE* operon, which is required for growth in arginine as a sole nitrogen source [56,57]. Arginine is an important amino acid involved in protein synthesis and metabolic gene expression, and modulates cellular and bacterial responses to environmental stressors [58,59]. Furthermore, arginine plays an important role in impacting the intestinal microbiome and activating intestinal innate immunity [60]. In colitis mouse models, increased dietary arginine reduced colitis and was thought to be related to the restoration of microbial diversity by increasing the relative abundance of *Bacteroidetes* [61]. Similarly, in non-tuberculous mycobacterial pulmonary disease murine models, the oral administration of arginine led to an increased abundance of *Bifidobacterium* species in the gut, boosting pulmonary immune defenses through the gut–lung axis [62]. Furthermore, the virulence of pathogenic bacteria such as Enterohemorrhagic *E. coli* (EHEC) is known to be modulated by arginine [63]. These bacteria carry a type III secretion system (T3SS), which is nanomachinery that can translocate effectors into the host cells, leading to colonization and disease [64]. In the presence of arginine, ArgR directly activates the expression of genes that encode T3SS, promoting the virulence of EHEC [63]. Although well-demonstrated in other pathogenic enteric bacteria, the role of arginine on *C. difficile* toxin production remains controversial and inconclusive. One study found that arginine could influence toxin production in *C. difficile*, where the addition of arginine enhanced toxin production in complex media [65]. Another study found that arginine had no effect on toxin production [66], yet a third study showed that an arginine insufficiency led to poor growth, but enhanced toxin production [67]. These findings, although inconsistent, highlight the importance of arginine regulation and sensing in the vulnerability of hosts to certain pathogens.

Although not previously explored in CDI, a number of the isolated KO groups have been explored in other pathogenic models, especially intestinal dysbiosis. Our in silico analyses identified a decrease in KO group K02440 (glycerol uptake facilitator protein), which plays an important role in glycerol diffusion in the cell. Although the importance of glycerol uptake facilitator protein in promoting bacterial pathogenesis is undetermined, glycerol is thought to play an important role in driving intestinal dysbiosis [68]. Glycerol-containing probiotics have also been explored as a potential therapy for CDI. Spinler et al. [69] found that human-derived *Lactobacillus reuteri* co-delivered with glycerol was effective against *C. difficile* colonization, but ineffective when treated with either *L. reuteri* or glycerol alone. Nevertheless, it was difficult to determine the contribution of the glycerol uptake facilitator protein in the success of FMT as we did not observe a change in the other enzymes that constitute the glycerol metabolic pathway such as glycerol-3-phosphate dehydrogenase.

Similarly, iron concentrations in the gut can influence the composition of intestinal microbiota. Iron is an essential element for all organisms and is linked to many cellular processes such as the transport and storage of oxygen, hormone synthesis, DNA replication, electron transfer, nitrogen fixation, and control of the cell cycle [70,71]. In humans, iron deficiencies have been implicated in a number of disease states, most commonly iron deficiency anemia [72]. As a strategy to improve iron intake, many supplements are fortified with iron; however, the absorption of iron is typically low, leading to large amounts of unabsorbed iron passing into the colon [73]. Although iron oxidation and utilization can be identified across different bacterial phyla, the largest abundance of iron oxidizers belong to *Proteobacteria* [74]. Many pathogenic enteric bacteria (i.e., *Salmonella*, *Shigella*, and *E. coli*) compete for unabsorbed dietary iron, which largely impacts the growth and virulence of these species [75]. A few protective bacteria such as *Lactobacilli* have profound metabolic capabilities, allowing them to successfully compete with iron-dependent bacteria in iron-rich environments, but also grow well in relative iron depletion [76]. Therefore, changes in the presence of unabsorbed iron in the gut can have a major influence on modulating intestinal microbiota. In a study on Kenyan infants, supplementation with iron-fortified powders promoted intestinal dysbiosis, reducing the abundance of *Bifidobacteria* and increasing the abundance of *Enterobacteriaceae* (i.e., *E. coli* and *Shigella*) and *Clostridium* species [77]. Similar findings have been demonstrated in other studies assessing the impact of iron-fortified foods on intestinal microbiota [78,79]. Interestingly, a study conducted on young women in South India found that women with an iron deficiency had a lower relative abundance of *Lactobacillus acidophilus* in the gut [79]. Although studies have been limited, iron homeostasis is important in modulating intestinal microbiota where both an iron deficiency and an iron excess can lead to changes in the composition of intestinal microbiota. In addition to influencing intestinal dysbiosis, iron homeostasis can also alter the concentration of SCFAs. For example, lower levels of propionate and butyrate were observed in rats with low levels of luminal iron, which is potentially a consequence of an associated gut dysbiosis [80]. When specifically exploring the impact of iron excess on CDI, Yamaki et al. [81] demonstrated that the growth of *C. difficile* and toxin production were both significantly increased in the presence of an iron excess. Furthermore, excess iron increased the minimum inhibitory concentration (MIC) of the antimicrobials used to commonly treat CDI, metronidazole and fidaxomicin, whilst the MIC of vancomycin was relatively unchanged [81]. Our analysis uncovered a decrease in ABC transporter groups involved in iron transport; namely, K02016 (iron complex transport system substrate-binding protein) and K02013 (iron complex transport system ATP-binding protein [EC:3.6.3.34]). These proteins play important roles in facilitating the uptake of iron from the periplasm to the cytoplasm of a variety of bacterial species [82]. The observed decrease in these two KO groups post-FMT may have been related to the restored diversity of the gut microbiota and reduced relative abundance of pathobionts that preferentially utilize iron for growth and virulence. Interestingly, although identified in our study, we did not observe significant changes in other KO groups that play important roles in bacterial iron transport such as ferrous iron transport protein A and ferrous iron transport protein B. Nevertheless, as iron homeostasis plays an important role in impacting bacterial virulence and growth as well as modulating intestinal microbiota, our findings prompt the need for further studies to evaluate how iron metabolism may contribute to FMT efficacy.

Several other KO groups that we identified also play a role in bacterial virulence and growth in non-CDI models; namely, K02528 (dimethyladenosine transferase), K04047 (starvation-inducible DNA-binding protein), and K01439 (succinyl-diaminopimelate desuccinylase [EC:3.5.1.18]). Although we demonstrated a significant change post-FMT, most KO groups had been identified in many bacterial phyla, including *Bacteroidetes*, *Firmicutes*, and *Proteobacteria*. Due to the non-specific expression of the KO groups, the implication of the change in KO groups post-FMT in either the prevention or promotion of RCDI remains unclear. Furthermore, alternative pathways may contribute to the production of metabolites associated with certain KO groups, suggesting that our functional profiling of the microbiome did not necessarily mirror the metabolic landscape of the gut. Thus, more holistic approaches are required to fully map the factors that promote the success of FMT. In this regard, our study identified several key microbial pathways that were altered during FMT in RCDI, which is key to engineering microbial consortia with a better therapeutic potential. 

## 5. Limitations

Although our study presented interesting findings of the potential changes in KO groups after FMT, hypothesizing the mechanisms of action of FMT in the treatment of RCDI, there were limitations to our study. First, this study was purely exploratory and inferred a gene presence based on fecal shotgun metagenomic sequencing data as we did not perform metabolomics to correlate the results. Second, shotgun metagenomics data and KEGG orthology groups are only as good as the available databases. Therefore, there are likely to be many other important metabolites and pathways that were missed with our in silico approach. Furthermore, the metagenomic sequencing was performed with shallow sequencing and this further limited what could be identified. For example, bacterial genes, bile salt hydrolase, and 7-α-dehydroxylase, which are important in RCDI and FMT, were not identified in the shotgun metagenomic sequencing results. Third, our patient population (*n* = 18) was small, thus prone to type 1 and type 2 errors. Finally, due to the high efficacy rates of FMT in the treatment of RCDI, all the patients included in this study were successfully treated for FMT, which means that we cannot comment if these changes would also be absent in patients with a failed FMT treatment.

To address these limitations, we suggest future studies to include a larger patient cohort and to include metabolomics analyses. It would also be important to include both successful and failed RCDI cases following FMT. Animal models can also be considered for further validation work.

## Figures and Tables

**Figure 1 microorganisms-11-01078-f001:**
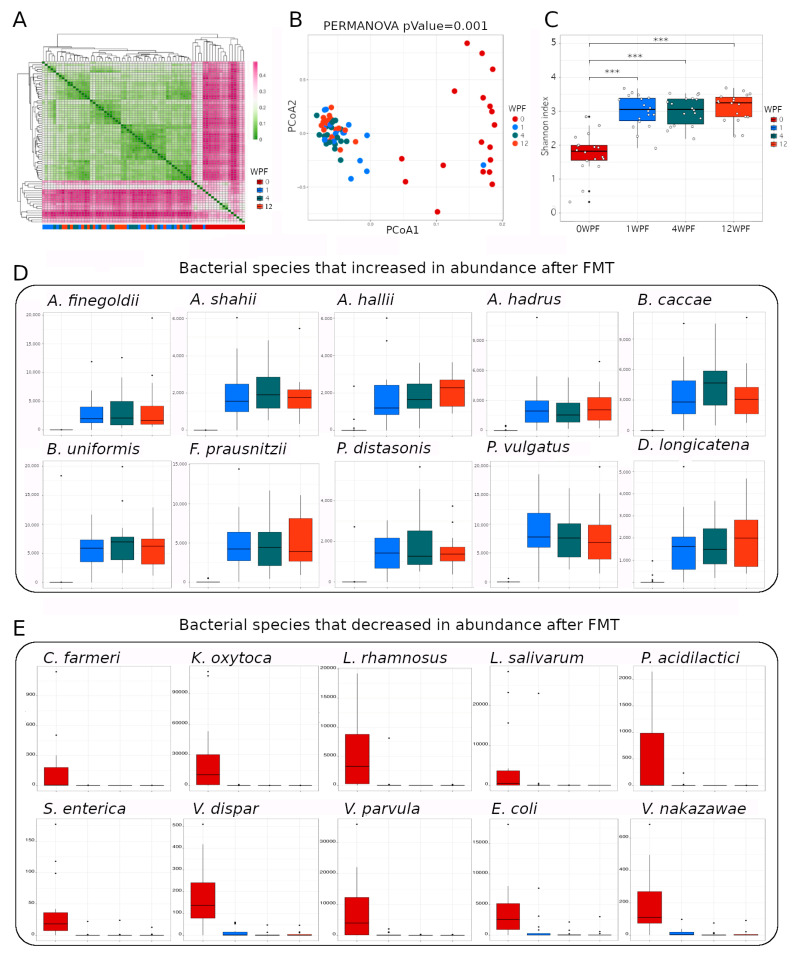
Fecal microbial transplantation (FMT) increases microbiome alpha diversity, increases the abundance of homeostatic bacteria, and decreases the abundance of opportunistic potentially pathogenic bacteria. (**A**) Hierarchical clustering on Bray–Curtis distances between all pairs of samples included in this study. (**B**) Principal coordinate analysis on Bray–Curtis distances between all pairs of samples included in this study. (**C**) Shannon diversity indices for each sample. In (**A**–**C**), 0 WPF, 1 WPF, 4 WPF, and 12 WPF refers to weeks post-FML. Hollow circles represent the Shannon index of individuals, while solid dots represent additional marks for observations considered outliers. *** refers to corrected *p* values < 0.001. (**D**) Ten representative species that were found increased after FMT (*A. finegoldii: Alistipes finegoldii*; *A. shahii: Alistipes shahii*; *A. hallii: Anaerobutyricum hallii*; *A. hadrus: Anaerostipes hadrus*; *B. caccae: Bacteroides caccae*; *B. uniformis: Bacteroides uniformis*; *F. prausnitzii: Faecalibacterium prausnitzii*; *P. distasonis: Parabacteroides distasonis*; *P. vulgatus: Phocaeicola vulgatus*; *D. longicatena: Dorea longicatena*). (**E**) Ten representative species that were found decreased after FMT (*C. farmeri: Citrobacter farmeri*; *K. oxytoca: Klebsiella oxytoca*; *L. rhamnosus: Lacticaseibacillus rhamnosus*; *L. salivarius: Ligilactobacillus salivarius*; *P. acidilactici: Pediococcus acidilactici*; *S. enterica: Salmonella enterica*; *V. dispar: Vaillonella dispar*; *V. parvula: Veillonella parvula*; *E. coli: Escherichia coli*; *V. nakazawae: Veillonella nakazawae*).

**Table 1 microorganisms-11-01078-t001:** KO Groups that increased post-FMT.

Cellular Process	KO Group	Function	*p*-Value	Time (95% CI)
DNA Replication	K02315: DNA replication protein DnaC	Accessory protein that facilitates the interaction of DnaB with single-stranded DNA/duplex DNA to aid in DNA replication [29]	1.95 × 10^−8^	6.64 (2.89 to 10.39)
Cellular Signaling	K07646: two-component system, OmpR family and sensor histidine kinase KdpD [EC:2.7.13.3]	Senses environmental signals; regulatory role in potassium transport in combination with KdpE; potassium homeostasis [30]	7.21 × 10^−4^	2.45 (1.09 to 3.80)

**Table 2 microorganisms-11-01078-t002:** KO Groups that decreased post-FMT.

Cellular Process	KO Group	Function	*p*-Value	Time (95% CI)
Metabolism	K04042: bifunctional UDP-N-acetylglucosamine pyrophosphorylase/glucosamine-1-phosphate N-acetyltransferase [EC:2.7.7.23, 2.3.1.157]	Trimeric bifunctional enzyme responsible for CoA-dependent acetylation of Glc-1-PO(4) to GlcNAc-1-PO(4) and catalyzes uridyl transfer from UTP to GlcNAc-1-PO(4) to form the final products UDP-GlcNAc and pyrophosphate [31]	1.28 × 10^−4^	−4.38 (−6.49 to −2.27)
	K01652: acetolactate synthase I/II/III large subunit [EC:2.2.1.6]	Catalyzes the conversion of 2 pyruvate molecules into acetolactate in the first common step to synthesize branch-chained amino acids (leucine, valine, and isoleucine) [32]	0.000513	−6.01 (−9.24 to −2.78)
	K02495: oxygen-independent coproporphyrinogen III oxidase [EC:1.3.99.22]	Catalyzes the oxygen-independent conversion of coproporphyrinogen-III to protoporphyrinogen-IX [33]	0.000279	−4.48 (−6.77 to −2.19)
	K03402: transcriptional regulator of arginine metabolism	Regulates arginine metabolism through negative regulation of operons involved in arginine biosynthesis [34]	3.93 × 10^−5^	−6.20 (−8.96 to −3.43)
	K02536: UDP-3-O-[3-hydroxymyristoyl] glucosamine N-acyltransferase [EC:2.3.1.191]	First enzyme involved in the formation of lipid A; lipopolysaccharide biosynthesis [35]	4.39 × 10^−4^	−3.69 (−5.65 to −1.73)
	K08591: glycerol-3-phosphate acyltransferase PlsY [EC:2.3.1.15]	Glycerophospholipid metabolism; catalyzes the transfer of an acyl group for acyl phosphate to glycerol-3-phosphate to form lysophosphatidic acid [36]	1.20 × 10^−5^	−4.25 (−6.02 to −2.49)
	K00656: formate C-acetyltransferase [EC:2.3.1.54]	Pyruvate metabolism, propionate metabolism, and butanoate metabolism; conversion of formate + CoA to pyruvate + CoA [37]	3.04 × 10^−4^	−4.98 (−7.54 to −2.41)
	K01207: beta-N-acetylhexosaminidase [EC:3.2.1.52]	Lysosomal isoenzyme that releases N-acetylglucosamine and N-acetylgalactosamine from glycoproteins, glycolipids, and glycosaminoglycans [38]	1.77 × 10^−4^	−2.87 (−4.29 to −1.45)
	K01439: succinyl-diaminopimelate desuccinylase [EC:3.5.1.18]	Catalyzes the hydrolysis of N-succinyl-L-diaminopimelic acid to produce L-diaminopimelic acid and succinate [39]	1.46 × 10^−4^	−3.92 (−5.83 to −2.01)
DNA Synthesis /Replication/Repair	K02313: chromosomal replication initiator protein	ATP-dependent; binds to origin of replication (oriC) to initiate the formation of DNA replication initiation complex; key in initiating and regulating chromosomal replication [40]	8.96 × 10^−5^	−3.85 (−5.67 to −2.04)
	K01756: adenylosuccinate lyase [EC:4.3.2.2]	Purine biosynthesis; catalyzes the conversion of SAICA ribotide into aminoimidazole carboxamide ribotide and conversion of succinyladenosine monophosphate to adenosine monophosphate [41]	1.05 × 10^−4^	−3.93 (−5.80 to −2.06)
	K02528: dimethyladenosine transferase	Ribosomal maturation; DNA mismatch repair [42]	2.22 × 10^−4^	−4.10 (−6.15 to −2.04)
	K04047: starvation-inducible DNA-binding protein	Protects bacteria against stressors (starvation, oxidative stress, metal toxicity, or thermal stress) through DNA binding and ferroxidase activity [43]	3.63 × 10^−5^	−4.77 (−6.88 to −2.65)
	K00526: ribonucleoside-diphosphate reductase beta chain [EC:1.17.4.1]	Catalyzes the biosynthesis of deoxyribonucleotides [44]	1.79 × 10^−4^	−4.91 (−7.34 to −2.48)
	K01524: exopolyphosphatase/guanosine-5′-triphosphate,3′-diphosphate pyrophosphatase [EC:3.6.1.11 3.6.1.40]	Catalyzes the conversion of guanosine 3′-diphosphate 5′-triphosphate + H_2_O to guanosine 3′,5′-bis(diphosphate) + H^+^ + phosphate [45]	4.48 × 10^−5^	−4.13 (−5.98 to −2.27)
	K06901: putative MFS transporter, AGZA family, xanthine/uracil permease	Pyrimidine metabolism [46]	3.60 × 10^−5^	−9.13 (−13.18 to −5.08)
Cellular Signaling	K06207: GTP-binding protein	Signaling protein; binds GTP [47]	5.27 × 10^−4^	−3.25 (−5.00 to −1.50)
Substrate Transport	K02440: glycerol uptake facilitator protein	Glycerol diffusion [48]	3.10 × 10^−4^	−5.70 (−8.64 to −2.76)
	K02016: iron complex transport system substrate-binding protein	Part of ATP-binding cassette (ABC) family of transporters; iron transport [49]	0.000359	−6.65 (−10.11 to −3.18)
	K10441: ribose transport system ATP-binding protein [EC:3.6.3.17]	Part of ABC family of transporters; ribose import [50]	3.11 × 10^−5^	−5.34 (−7.69 to −2.99)
	K02761: PTS system, cellobiose-specific IIC component	Recognizes and binds sugars and transports them across the cell membrane into the cytoplasm [51]	1.37 × 10^−4^	−11.12 (−16.52 to −5.73)
	K02013: iron complex transport system ATP-binding protein [EC:3.6.3.34]	Part of ABC family of transporters; iron transport [49]	6.03 × 10^−5^	−8.61 (−12.56 to −4.66)
	K02073: D-methionine transport system substrate-binding protein	D-methionine uptake [52]	4.28 × 10^−5^	−5.92 (−8.57 to −3.26)
Miscellaneous	K00425: cytochrome d ubiquinol oxidase subunit I [EC:1.10.3.14]	Electron/proton transport and part of aerobic respiratory chain; oxidative phosphorylation [53]	2.46 × 10^−4^	−4.32 (−6.51 to −2.13)

## Data Availability

No new data were created.

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
