# Peer review of "In Silico Analysis of Changes in Predicted Metabolic Capabilities of Intestinal Microbiota after Fecal Microbial Transplantation for Treatment of Recurrent Clostridioides difficile Infection"

_microorganisms, 2023, doi:10.3390/microorganisms11041078_

Round 1

Reviewer 1 Report

Major concerns:

The article has severe flaws in methods, and all the bioinformatic analyses need to be included in the text. All standard analyses for microbiome data or microbial ecology were not performed. Besides, the author's claims need to be clarified. All the manuscript refers to some picked functions correlated with time, most of which are housekeeping functions.

Minor concerns:

Line 49. There is a misconception about the statement "microbial ecology is restored". Microbial Ecology is a branch of the Ecology and Microbiology fields, so a scientific discipline, not a characteristic of the intestine that can be restored. Please reformulate this sentence.

The entire bioinformatic methods still need to be included—the sequences cleaning, normalization, annotation, processing, etc.

Please rewrite the sentences in Results, lines 160-162, which are hard to understand.

All the results about the metagenome libraries need to be included (total number of reads, library size, metagenomic coverage, etc.). It must be included in the results.

Line 171. The significant p-value looks random. The authors must specify a standard p-value threshold based on the confidence interval (0.05, 0.01, or 0.001).

Author Response

Major concerns:

The article has severe flaws in methods, and all the bioinformatic analyses need to be included in the text. All standard analyses for microbiome data or microbial ecology were not performed. Besides, the author's claims need to be clarified. All the manuscript refers to some picked functions correlated with time, most of which are housekeeping functions.

We thank you for your excellent suggestions.

In response to your major concerns, we have added further details around the bioinformatics methods from JAMA and further results of the metagenome libraries have also been included (Figure 1).

Furthermore, we recognize the KO groups identified in our study are involved in housekeeping functions; we have acknowledged this in our discussion section (lines 336-340).

Minor concerns:

Line 49. There is a misconception about the statement "microbial ecology is restored". Microbial Ecology is a branch of the Ecology and Microbiology fields, so a scientific discipline, not a characteristic of the intestine that can be restored. Please reformulate this sentence.

We agree the term “microbial ecology” is unclear. We have changed this to microbial diversity.

The entire bioinformatic methods still need to be included—the sequences cleaning, normalization, annotation, processing, etc. Methodology from JAMA

We apologize for the omission. This has now been included in the methods.

Please rewrite the sentences in Results, lines 160-162, which are hard to understand.

This has been rewritten (lines 274-277).

All the results about the metagenome libraries need to be included (total number of reads, library size, metagenomic coverage, etc.). It must be included in the results.

We have generated a figure (Figure 1) that includes hierarchical clustering and principal coordinate analyses of Bray-Curtis distances, which both indicate that the taxonomic profile of subjects changed dramatically from pre to post FMT. We also present the Shannon diversity indices and Wilcoxon test comparing time point 0 (pre FMT) to  1, 4 or 12 weeks post FMT. After correction of p Values, all comparison were highly significant. Finally, we present the normalized abundance of bacterial species whose abundance dramatically increased or decreased after FMT, consistent with a shift towards a more homeostatic microbiome.

Line 171. The significant p-value looks random. The authors must specify a standard p-value threshold based on the confidence interval (0.05, 0.01, or 0.001).

Thank you for this suggestion. We have corrected the significant p-value to reflect a standard p-value threshold. This is explained in lines 262-266.

Reviewer 2 Report

Some recent relevant publications might be added to the reference list, including Oleskin, A. V. and Shenderov, B. A. (2020). MICROBIAL COMMUNICATION AND MICROBIOTA-HOST INTERACTIONS: BIOMEDICAL, BIOTECHNOLOGICAL, AND BIOPOLITICAL IMPLICATIONS.. Nova Science Publ.: Hauppauge, New York. or Shenderov, B. A. (2016). The microbiota as an epigenetic control mechanism. Chapter 11. In: Nibali. L, & Henderson, B. (Eds.). The Human Microbiota and Chronic Disease: Dysbiosis as a Cause of Human Pathology. 1st ed, J. Wiley & Sons, pp.179-197

Author Response

Some recent relevant publications might be added to the reference list, including Oleskin, A.V. and Shenderov, B.A. (2020). MICROBIAL COMMUNICATION AND MICROBIOTA-HOST INTERACTIONS: BIOMEDICAL, BIOTECHNOLOGICAL, AND BIOPOLITICAL IMPLICATIONS.. Nova Science Publ.: Hauppauge, New York. or Shenderov, B. A. (2016). The microbiota as an epigenetic control mechanism. Chapter 11. In: Nibali. L, & Henderson, B. (Eds.). The Human Microbiota and Chronic Disease: Dysbiosis as a Cause of Human Pathology. 1st ed, J. Wiley & Sons, pp.179-197

Thank you for these excellent suggested publications. We have added these publications into our introduction (references 20, 21).